# The Promising Role of a Zebrafish Model Employed in Neural Regeneration Following a Spinal Cord Injury

**DOI:** 10.3390/ijms241813938

**Published:** 2023-09-11

**Authors:** Chih-Wei Zeng, Huai-Jen Tsai

**Affiliations:** 1Department of Molecular Biology, University of Texas Southwestern Medical Center, Dallas, TX 75390, USA; chih-wei.zeng@utsouthwestern.edu; 2Hamon Center for Regenerative Science and Medicine, University of Texas Southwestern Medical Center, Dallas, TX 75390, USA; 3Department of Life Science, Fu Jen Catholic University, New Taipei City 242062, Taiwan

**Keywords:** spinal cord injury, cell transplantation therapy, neural regeneration, immune response, glial scars, zebrafish, progenitor cells

## Abstract

Spinal cord injury (SCI) is a devastating event that results in a wide range of physical impairments and disabilities. Despite the advances in our understanding of the biological response to injured tissue, no effective treatments are available for SCIs at present. Some studies have addressed this issue by exploring the potential of cell transplantation therapy. However, because of the abnormal microenvironment in injured tissue, the survival rate of transplanted cells is often low, thus limiting the efficacy of such treatments. Many studies have attempted to overcome these obstacles using a variety of cell types and animal models. Recent studies have shown the utility of zebrafish as a model of neural regeneration following SCIs, including the proliferation and migration of various cell types and the involvement of various progenitor cells. In this review, we discuss some of the current challenges in SCI research, including the accurate identification of cell types involved in neural regeneration, the adverse microenvironment created by SCIs, attenuated immune responses that inhibit nerve regeneration, and glial scar formation that prevents axonal regeneration. More in-depth studies are needed to fully understand the neural regeneration mechanisms, proteins, and signaling pathways involved in the complex interactions between the SCI microenvironment and transplanted cells in non-mammals, particularly in the zebrafish model, which could, in turn, lead to new therapeutic approaches to treat SCIs in humans and other mammals.

## 1. Introduction

Spinal cord injury (SCI) typically results in axonal damage and the death of neurons and glial cells. The secondary injury phase is primarily caused by uncontrolled inflammation, excitotoxicity, edema, ischemia, and chronic demyelination at the injury site. Glial scars generated at the wound site can inhibit axonal regeneration [1]. Each year, an estimated 250,000 to 500,000 individuals worldwide suffer from SCIs [2,3], which are usually severe and life-threatening, with approximately half of patients dying [4,5]. The spontaneous repair of neural cells following an SCI is very low, and current treatment strategies mostly rely on mechanical decompression, symptomatic relief, supportive care, and early rehabilitation. With the development of stem cell technology, cell-based transplantation using neural stem cells (NSCs) or induced pluripotent stem cells (iPSCs) with differentiation potential has been considered a promising therapeutic approach. For example, Maeda et al. (2021) transplanted bone marrow mesenchymal stem cells into the SCI lesions in rats, resulting in better functional recovery [6]. Other stem cells currently available for transplantation include embryonic stem cells [7], umbilical cord mesenchymal stem cells [8], and Schwann cell preparations [9]. Furthermore, Ito et al. (2021) showed that transplanted LOTUS-overexpressing hiPSC-NS/PCs could enhance the functional recovery of motor impairment, indicating the possible clinical application of stem cell-based transplantation for the treatment of SCIs [10]. These findings highlight the potential of stem cell-based transplantation as a promising therapeutic approach for the treatment of SCIs.

The spontaneous regeneration of axons in the central nervous system (CNS) of mammals is limited after injury. This can be attributed to both decreased the intrinsic growth capacity of mature neurons during development and various environmental factors [11,12]. However, these challenges have been successively addressed in mammalian studies over the years by many researchers. For example, one recent study reported the formation of neurospheres from neural precursor cells cultured from the adult human spinal cord by the fourth day. After ten days, these neurospheres can differentiate into astrocytes and neurons [13]. Following focal ischemia or middle cerebral artery occlusion in rats, cells located in the subventricular zone (SVZ) undergo proliferation and differentiation into neuroblasts, which subsequently migrate to the damaged area [14,15]. In addition, an early study by Richardson et al. (1980) demonstrated that central neurons have the capacity to regenerate when the environment around the injury is favorable [16]. Building on these pioneering studies, various approaches have been used to promote the growth of injured axon through the injury area. This suggests that local neural cell death caused by ischemia or hypoxia induces the proliferation of cells in the SVZ, which then differentiate into neural cells and migrate to the damaged area for repair. This promising discovery was, however, dampened by the finding that hypoxia can stimulate mitochondria to produce reactive oxygen species (ROS), which ultimately results in oxidative stress-induced neural cell apoptosis [17] to which the CNS is particularly susceptible [18,19]. Nevertheless, when newborn piglet brains are subjected to a period of hypoxia followed by oxygen, Ara et al. (2013) showed an increase in neural stem progenitor cells in the SVZ, along with the appearance of new neural cells and many as-yet uncharacterized cell populations that promote proliferation in the striatum and white matter [20]. Collectively, therefore, these studies suggest that a group of mammalian cells may differentiate into neural cells and participate in neural repair. Moreover, these heterogeneous cell populations and corresponding regenerative mechanisms remain to be elucidated. Importantly, clinical studies on human subjects have also begun to corroborate the therapeutic promise of stem cell-based transplantation in SCIs. For example, phase I/II trials conducted by Satti et al. (2016) explored the safety and feasibility of transplanting autologous bone marrow-derived mesenchymal stem cells in patients with chronic SCIs, showing improvements in motor function and sensation [21]. Tabakow et al. (2013) demonstrated the potential of using olfactory ensheathing cells (OECs) for SCI treatment for patients exhibiting enhanced neurological functions after transplantation [22]. Furthermore, Liu et al. (2022) showed that combining neural stem cells (NSCs) with a bioengineered scaffold may represent a promising therapeutic strategy for SCI repair [23]. These human studies add a critical dimension to our understanding of the gap between preclinical animal studies and human clinical trials, thus highlighting the translation of stem cell technologies from bench to bedside. The collective findings underscore the potential of stem cell-based transplantation as an evolving and promising therapeutic approach for SCIs, revealing new horizons for future research and clinical practices. 

Microglial cells in the CNS initiate an inflammatory response following SCIs in mammals, thereby attracting other immune cells to the damaged area and accelerating the immune response [24,25]. However, reactive astrocytes further contribute to scarring in the damaged area by expressing chondroitin sulfate proteoglycans [26,27], a process that can inhibit axonal regeneration and outgrowth [28,29]. To counteract this outcome, researchers have employed two key strategies: the use of chondroitinase ABC, which breaks down inhibitory components within the glial scar [30,31], and the transplantation of ensheathing glial cells from the olfactory bulb or olfactory mucosa, thereby creating “channels” that allow regenerating axons to pass [32]. These innovative approaches specifically target the inhibitory effects of glial scarring, paving the way for promising advancements in SCI recovery and neural repair.

In non-mammals, such as amphibians, birds, and fish, the ability to regenerate neurons in any area of the brain and spinal cord can occur [33,34,35]. Nerve regeneration takes place in the ventricular zone [36,37,38,39,40]. Neural regeneration capabilities exhibit remarkable diversity, reflecting different underlying genetic and cellular mechanisms.

### 1.1. Lampreys

Lampreys, jawless vertebrates, have been the focus of a study on spinal cord regeneration due to their capacity to fully recover their locomotor functions following a spinal cord transection [41,42], even though spinal cord axons do not have myelin sheaths [43]. Recent research has shed light on various cellular and molecular mechanisms involved in their regenerative abilities, including the activation of specific signaling pathways and the role of glial cells [44]. These studies are inspiring new avenues for therapeutic strategies in mammals since spinal cord injuries typically result in permanent loss of function.

### 1.2. Salamanders (Axolotls)

Axolotls are famous for their regenerative abilities. Genes, such as *sp9* and *msx1*, are crucial to understanding limb and spinal cord regeneration. Unlike lampreys, axolotls deploy macrophages that express both inflammatory and anti-inflammatory cytokines, facilitating regeneration without scarring [45,46].

### 1.3. Tadpoles (Xenopus)

In Xenopus tadpoles, SCI in the tail induces tail regeneration [47], along with the restoration of swimming ability, and the immune response induced by the SCI is related to neural regeneration [48]. SCI induces tail regeneration where Sox2-positive cells play a key role. Genes, such as *Notch1*, have been explored for their role in increased regenerative capacity during development in comparison to the decreased regenerative capacity observed during metamorphosis [49].

### 1.4. Zebrafish

Zebrafish, on the other hand, has strong neural regeneration abilities from the embryonic to adult stages. After a complete transection [50] or mechanical crush injury [51] on the spinal cord, newly developed motor neurons are generated, resulting in restoring their normal swimming activity. They can restore their normal swimming ability post-injury as a result of their robust regenerative capacities [52]. Furthermore, recent studies have revealed a coordinated action of immune-related genes that support neuronal regeneration without hindering growth, highlighting the balance between immune response and regeneration [53].

### 1.5. Birds (Canaries)

Some birds, such as canaries, exhibit seasonal neurogenesis, particularly in regions of the brain responsible for song production. This unique phenomenon has been linked to specific behaviors, such as song learning and adaptation. Studies have identified key molecules, such as the brain-derived neurotrophic factor (BDNF), which play a vital role in these processes [54]. BDNF expression has been correlated with changes in neural connectivity and regeneration within the song control system, reflecting the functional plasticity of the canary’s brain.

In summary, the ability to regenerate neurons varies widely across non-mammalian species, and distinct genetic pathways underlie these differences. Lampreys lack glial scarring, facilitating regeneration, while axolotls use a balanced immune response. Tadpoles offer insights into the developmental changes in regenerative capacity, and zebrafish demonstrate a robust, orchestrated genetic response. Birds, such as canaries, provide unique examples of behavior-linked regeneration. These variations in genetic and cellular responses offer valuable models for exploring potential therapeutic approaches to human SCIs, contributing to our growing understanding of neural regeneration across the animal kingdom.

## 2. Current Challenges in the Treatment of SCIs

No effective treatments are clinically available for patients with SCIs, largely owing to the many unknowns. Nonetheless, many studies reporting on SCIs have led to an in-depth understanding of the biological reactions that occur in SCI-injured tissue. As a result, some treatment options have been proposed and explored in clinical settings. One of the notable successes in this area has been the transplantation of olfactory ensheathing glial cells together with peripheral nerve grafts. This treatment showed promising results in a patient with an incomplete SCI [55]. It represents a significant breakthrough and a practical application of the experimental findings to human patients. Other efforts include a study by Sabapathy et al. (2015) who explored cell transplantation in the treatment of SCIs in rats, mice, and humans [56]. They discovered the development of a microenvironment imbalance at the damage site, which, in turn, resulted in low cell survival rates. However, since the dose and proportion of transferred cells are not well understood and optimized, cell transplantation for nerve regeneration has not met the expectations well. Therefore, many issues, such as those raised by Sabapathy and coworkers, remain to be addressed for successful stem cell transplantation following an SCI. Accordingly, we described a concise visual summary of the main challenges in SCI treatment and highlighted below four points representing key bottlenecks in SCI research, helping to better understand the complexity of problems and various factors needed to be comprehensively addressed for successful therapeutic interventions (Figure 1).

### 2.1. Precise Cell Types Involved in the Process of Nerve Regeneration Are Still Unknown

At present, stem cell transplantation therapies for SCIs often use a single stem cell type, such as adult bone marrow stem cells, hematopoietic stem cells, and mesenchymal stem cells (MSCs [57]). For example, Sasaki et al. (2009) transplanted bone marrow cells from mice into the SCI lesion and observed a re-myelination of the nerve cells; however, the differentiated nerve cells were nonfunctional [58]. Other studies have shown the presence of unidentified cell populations involved in SCI repair in addition to NSCs or stem-like cells with a differentiation capacity [59,60]. Therefore, recent studies have reported mixed cell populations that may participate in nerve regeneration, including Schwann cells [61], olfactory ensheathing cells [62], and neurotrophin-expressing fibroblasts, such as BDNF and NT-3 [63]. At present, however, a definitive analysis of cell populations engaged in nerve regeneration is difficult for two reasons. First, cell populations that participate in nerve regeneration are identified using specific protein markers, including SOX2 for neural progenitors, A2B5 for astrocyte/glial progenitor cells, GFAP for astrocytes, and NG2 for oligodendrocyte progenitor cells and Schwann cell-like progenitor cells. Second, cell populations involved in nerve regeneration have been examined using only single-gene transgenic strains [64,65]. Therefore, since cell populations involved in nerve repair are heterogeneous, their identification cannot be achieved by using a single marker protein or a specific gene transgenic strain.

### 2.2. The Microenvironment and Cell Transplantation in SCIs

SCIs lead to a highly abnormal microenvironment at the injury site, characterized by the apoptosis or necrosis of surrounding tissues and cells, the production of ROS, and an acute inflammatory response [66]. Therefore, this hostile microenvironment poses challenges for cell transplantation strategies, as the transplanted cells may struggle to survive and function effectively under such conditions. For example, Himes et al. (2006) found that human MSCs transplanted to the SCI site in mice failed to induce recovery, likely owing to such complications as inflammation, glial scarring, hypoxia, and stress-labile cell types [67]. However, it should be noted that not all studies have reported poor cell survival post-transplantation. For example, Xue et al. (2020) demonstrated that neural stem cells preconditioned with growth factors did exhibit significantly higher survival rates when transplanted into an SCI model compared to non-preconditioned cells [68]. 

In fact, a body of evidence suggests that transplanted cells can survive, even thrive, in this adverse microenvironment. Studies using olfactory ensheathing cells [69], Schwann cells [70], and even mesenchymal stem cells preconditioned with anti-inflammatory agents have shown robust survival and functional integration post-transplantation in SCI models [71]. These findings indicate that the adverse microenvironment conditions of SCI-damaged tissue are not universally prohibitive for cell transplantation and that strategies may be developed to enhance cell survival and function. Moreover, the variability in outcomes can be attributed to multiple factors. For instance, a study by Matyas et al. (2017) demonstrated that the use of adjunctive treatments significantly improved the survival and functionality of transplanted cells [72]. Therefore, strategies aimed at either modifying the adverse microenvironment at the injury site or identifying particular stem cell types that can withstand these conditions are crucial for advancing SCI treatment. Ra et al. (2011) demonstrated the safety of autologous adipose-derived mesenchymal stem cells (AD-MSCs) via intravenous infusion [73]. However, the study design did not include a control group, and other limitations were prohibitive of a positive outcome, such as the small sample size and short observation period, thus precluding a comprehensive clinical evaluation of AD-MSCs in SCI patients.

In summary, while the microenvironment at the SCI damage site presents significant challenges for cell transplantation, it is not an insurmountable barrier, as suggested by the evidence provided above, which holds out some hope of optimizing these strategies for clinical translation.

### 2.3. Inhibitory Immune Response Obstructs Nerve Regeneration Following an SCI

Following SCIs in mammals, immune cells of the CNS initiate a complex inflammatory response that can have both beneficial and harmful effects on nerve regeneration.

#### 2.3.1. Microglia

As resident immune cells of the CNS, microglia are often the first to respond to an injury. They can promote tissue repair by clearing cellular debris and releasing growth factors. However, chronic activation can lead to the release of pro-inflammatory mediators, contributing to secondary injury processes [74,75].

#### 2.3.2. Neutrophils

These immune cells are recruited early after injury and play a multifaceted role in the response to SCIs. On the one hand, they can release enzymes and ROS, aggravating damage to neural tissue. On the other hand, neutrophils can also facilitate wound healing and tissue repair in the early stages of injury. This dual role makes them a complex target for therapeutic interventions [76].

#### 2.3.3. Macrophages

Macrophages are immune cells that play a critical dual role in the response to SCIs. M1 macrophages promote inflammation and may exacerbate tissue damage, while M2 macrophages assist in tissue repair and regeneration [77,78]. The balance between these phenotypes can influence the outcome of the regeneration process, and understanding how this balance is achieved is key to developing therapeutic strategies.

#### 2.3.4. Lymphocytes

In addition to T and B cells, lymphocytes contribute to the immune response where some subsets promote inflammation, but others support tissue repair [79].

These immune cells release pro-inflammatory mediators, such as cytokines and chemokines, which can inhibit neuronal regeneration by causing cell death, demyelination, and the suppression of axonal regeneration [80]. Upregulated axonal growth inhibitors, such as Nogo-A [81], myelin-associated glycoprotein [82], and oligodendrocyte-myelin glycoprotein [83], further impede axonal regrowth. The recruitment and infiltration of peripheral immune cells into the SCI site can exacerbate inflammation, releasing factors that cause additional damage to the CNS [84]. Thus, immune responses following SCIs represent a delicate balance between beneficial repair mechanisms and detrimental inflammatory processes.

To overcome this challenge and promote nerve regeneration, an intensive investigation into therapeutic strategies targeting inhibitory immune responses is required. Such strategies may include anti-inflammatory agents, the neutralization of axonal growth inhibitors, the modulation of immune responses to create a more permissive environment for regeneration [85,86,87], and cell-based therapies, such as MSC transplantation [88,89]. However, translating these into clinical practice presents a number of challenges. These include ensuring an optimal dosage, administration routes, and long-term safety, as well as minimizing the side effects. Other challenges involve determining the best time for the intervention and exploring synergistic treatment combinations. It is also necessary to adapt the findings from animal models to human physiology, requiring further clinical trials that meet the regulatory standards and ethical guidelines, especially the trials involving stem cell therapies. Tailoring treatments to the individual patient’s needs and specific injuries, along with creating targeted delivery systems, should also be explored. The ongoing research seeks to overcome these challenges, aiming for effective SCI therapies. Collaborations across sectors are vital for real-world clinical translations.

### 2.4. Glial Scarring Impedes Axonal Regeneration after an SCI

In a mammalian CNS, astrocytes become reactive following an injury and form glial scars. This is a multifaceted process involving numerous signaling pathways and key molecules.

#### 2.4.1. Chondroitin Sulfate Proteoglycans (CSPGs)

Produced by astrocytes, CSPGs consist of a core protein linked to sulfated glycosaminoglycan side chains. CSPGs interact with receptors, such as NgR1, physically obstructing axonal growth and actively inhibiting regeneration. They also potentially protect against bacterial invasion but hinder functional recovery [90,91,92,93,94].

#### 2.4.2. Transforming Growth Factor-Beta (TGF-β) Signaling

TGF-β is a multifunctional cytokine that stimulates astrocyte activation and CSPG synthesis. It binds to specific receptors, initiating a cascade that promotes scarring and contributes to the inhibitory environment surrounding the injury [95].

#### 2.4.3. RhoA/ROCK Signaling Pathway

RhoA is a small GTPase, and its downstream effector, ROCK, regulates the cytoskeleton of growing axons. The activation of this pathway leads to growth cone collapse, stalling axonal regeneration. Inhibitors of RhoA/ROCK have shown potential in overcoming this inhibition [96].

#### 2.4.4. Integrin Signaling

Integrins, such as αvβ3 and αvβ8, modulate cellular adhesion and migration by interacting with extracellular matrix proteins, such as CSPGs. These interactions can influence scar dynamics and the regenerative response, making it a potential therapeutic target [97].

#### 2.4.5. Sema3A/NRP-1/PlexinA Signaling

Semaphorin 3A (Sema3A) is a chemorepellent molecule that binds to Neuropilin-1 (NRP-1) and PlexinA receptors, hindering axonal growth [98]. This complex signaling interaction contributes to the non-permissive environment of the glial scar, inhibiting regeneration [99,100]. Understanding how to modulate this signaling pathway may offer new strategies to promote recovery after SCIs.

#### 2.4.6. Stat3 Signaling

The signal transducer and activator of transcription 3 (Stat3) is involved in astrocyte reactivity. Its activation promotes the transcription of genes associated with glial scar formation, such as GFAP, further contributing to the inhibitory environment [101].

The research at present focuses on the therapeutic strategies that target these inhibitory pathways. For example, Chondroitinase ABC degrades CSPGs, improving axonal regeneration in animal models [102,103,104]. Pharmaceutical agents targeting RhoA/ROCK and other signaling pathways have also shown potential [105,106]. Additionally, cell transplantation strategies offer promising avenues to modulate the glial scar environment [107,108,109,110].

### 2.5. Difficulty Encountered in Bridging Large Gaps in the Spinal Cord after a Severe Injury

In cases of severe SCIs, large gaps or cavities can be formed at the injury site, which poses a significant challenge for axonal regeneration and the restoration of neural connections [111]. The lack of a supportive structure or scaffold within these gaps hinders the ability of regenerative axons to cross the lesion site to re-establish functional connections with their target neurons [112]. To address this issue, researchers are developing various biomaterial scaffolds and hydrogels that can be implanted into the lesion site to provide a supportive material for axonal growth and tissue repair [113,114,115,116]. The scaffolds can also be engineered to deliver growth factors or other bioactive molecules that promote axonal regeneration and cell survival [117]. This is often achieved through the incorporation of slow-release mechanisms within the scaffold, allowing for a sustained delivery of these molecules over time [118]. Some advanced scaffolds are being developed for use in conjunction with cell-based therapies. For instance, the transplantation of neural progenitor cells or Schwann cells into these scaffolds has been shown to further enhance the regeneration [119]. Furthermore, these scaffolds can be designed to mimic the native extracellular matrix, provide mechanical support, and deliver growth factors or other bioactive molecules that promote axonal regeneration and cell survival [120,121]. Moreover, some scaffolds can be combined with cell-based therapies, such as the transplantation of neural progenitor cells or Schwann cells, to further enhance their regenerative potential [122,123]. Despite the promising progress in the development of biomaterial scaffolds for SCI treatment, the challenges remain in optimizing the physical properties, biocompatibility and turnover rate of treated materials, as well as ensuring their long-term safety and efficacy in clinical settings. 

### 2.6. Limited Ability of Functional Recovery and Integration of Newly Regenerative Axons

Even when axonal regeneration is achieved, functional recovery remains a challenge due to the limited ability of newly regenerative axons to integrate into the existing neural circuits, reestablishing a well-organized and functional synaptic connection. The complexity of the spinal cord’s neural networks and the precise timing and organization of synaptic connection make it difficult for regenerative axons to establish an appropriate connection with their target neurons [124,125]. Furthermore, the loss of neuronal circuitry and synaptic plasticity after an SCI may lead to limited functional recovery, even when axonal regeneration is achieved [126]. To overcome this challenge, researchers are exploring various approaches to promote synaptic plasticity and enhance the integration of regenerative axons into the existing neural circuits. These strategies include electrical stimulation, optogenetic manipulation, and the administration of neuromodulatory agents in order to promote functional reorganization and plasticity within the spinal cord after an SCI [127,128,129]. However, again, further research is required to optimize these approaches and better understand the mechanisms underlying the functional recovery and synaptic integration of regenerative axons following SCIs.

## 3. Zebrafish Present an Opportunity to Study Neural Regeneration in SCIs

In addition to the studies on mammals and non-mammals mentioned above, many SCI studies currently use a zebrafish model system. An overview of the genetic tools used in zebrafish research to study neural regeneration after an SCI is included. It outlines the purpose of each tool and lists example studies that employ these techniques to investigate gene and protein functions in the context of neural regeneration (Table 1). For instance, Hui et al. (2010) found that a crush injury to a zebrafish spinal cord can induce nearby glial cells and other unknown cells to proliferate in the damaged area [130]. Moreover, new neurons appear near the crush site. A stab injury to the central dorsal telencephalon of zebrafish induces glial cells, NSCs, and unknown cells in the telencephalic ventricular zone to proliferate and eventually migrate to the damaged neurons [64,131]. In addition, recent studies have also utilized zebrafish to identify numerous candidate proteins involved in the process of neuroregeneration following an SCI. For example, Mokalled et al. (2016) identified the connective tissue growth factor a (CTGFa) as a potential candidate gene able to induce neuronal regeneration after an SCI [132]. Zeng et al. (2021) found that Caveolin 1 (Cav1), a membrane protein, was significantly upregulated in the rostral side of glial cells at the injury region and was responsible for axonal regrowth [51]. Lee et al. (2022) discovered that acidic nuclear phosphoprotein 32 family member A (ANP32a) played a positive role in the regeneration of zebrafish embryos with SCIs [133].

Another advantage of studying zebrafish is the availability of various transgenic lines that can be used to understand the complex cell types involved in neural regeneration during an SCI. For example, *Tg(her4.1:mCherryT2ACreERT2)* labels *her4.1*-positive ventricular radial glial progenitor cells [64], *Tg(−3.5dbx1a:egfp)* labels radial glial progenitor cells that differentiate into neurons during embryogenesis [65], and *Tg(−8.4ngn1:egfp)* labels young migrating NPCs [131]. However, Hui et al. (2015) reported that multiple progenitors, including SOX2-positive neural, A2B5-positive astrocyte/glial, NG2-positive oligodendrocyte, and Schwann cell-like progenitors, were also involved in neural repair, as well as other unknown cells [134]. Lee et al. (2011) generated a zebrafish transgenic line, *huORFZ*, which harbored an upstream open reading frame (uORF) of human *chop* mRNA (*huORF^chop^*) fused with GFP reporter and driven by cytomegalovirus promoter. This cassette was able to inhibit the translation of the downstream main coding sequence of *gfp* under a stress-free condition [135]. However, GFP is exclusively expressed in the CNS of *huORFZ* embryos in the presence of ER stress, such as heat-shock [135], hypoxia [35,40] and mechanical injury [51]. Moreover, when *huORFZ* embryos are exposed to hypoxic stress, GFP is expressed only in the specific population of subtype cells within the CNS, termed hypoxia-responsive recovering cells (HrRCs) [35]. HrRCs, consisting of various subtypes, contribute to neuronal regeneration post-hypoxia and play both rescue and regenerative roles in the post-lesion microenvironment [35]. Similarly, *huORFZ* embryos treated with mechanical SCIs revealed another specific population of subtypes, termed SCI stress-responsive regenerating cells (SrRCs). The major subtypes of SrRCs are radial glia (RGs-SrRCs) and neural stem/progenitor cells (NSPCs-SrRCs). They are highly resistant to SCI stress and are able to proliferate, differentiate, and migrate in order to play a crucial role in axonal regeneration through their collective complex of subtype cells [40]. As mentioned earlier, the complex cell groups involved in neural regeneration, even when using the zebrafish model, cannot be fully understood by solely relying on specific and specialized protein markers.

Nonetheless, the zebrafish model offered the opportunity to manipulate the expressions of known or unknown genes and proteins in a manner that could be applied to mammalian models of SCIs. For example, recent studies by Cui et al. (2021) showed that neuropeptide Y (NPY) expression in motor neurons promoted descending axonal regeneration and locomotor recovery in adult zebrafish after an SCI [136]. Mirchandani-Duque et al. (2022) expanded this work and identified NPYY1 and GAL2 receptors that mediated increased survival and neurite outgrowth in human and mammalian hippocampal neuronal cells [137]. As such, the zebrafish model, with its high regenerative potential, provides a valuable platform for the discovery of regenerative genes that may be latent in mammals, but can be activated in zebrafish to induce neurogenesis. Furthermore, researchers can also perform molecular biotechnologies on a zebrafish model, such as gene knockdown using antisense oligonucleotide morpholinos [138,139], knockout or knockin using the CRISPR/Cas9 system [140,141], overexpression using a microinjection of mRNA, and gene transfer using the AAV-ITR cassette [142] or Tol2 transposon-mediated transgenesis [143]. These genetic tools can help to identify novel genes and proteins that play essential roles in neural regeneration and repair after SCIs, as well as their functional interactions with other molecular pathways. Moreover, zebrafish displays conserved molecular and cellular processes with mammals [144], making them an attractive model for studying neural regeneration. The similarities in the gene expression profiles and molecular signaling pathways between zebrafish and mammals can facilitate the translation of the findings from zebrafish model to the mammalian system [145,146]. Thus, an in vivo study on a zebrafish model can contribute to the development of novel therapeutic approaches for promoting neural regeneration in mammals with SCIs. Overall, zebrafish studies have the potential to increase our understanding of neural regeneration during SCIs, including the proliferation and migration of various cell types and the involvement of various progenitor cells. However, to gain a full understanding of the molecular mechanisms, signaling pathways, proteins, and cell types engaged in neural regeneration in zebrafish, continued intensive study is required. In addition to the rich array of transgenic zebrafish lines that facilitate studies on neural regeneration, various genetic tools have also been employed in mammalian models of SCIs. For example, in zebrafish, Gal4-UAS and Cre-Lox systems are commonly used to manipulate gene expression, specifically in neurons or glial cells [147]. In the mammalian model, CRISPR/Cas9 was used to knock out inhibitory molecules, such as Nogo-A, showing promise for enhanced axonal growth post-injury [148,149]. Viral vectors have also been instrumental in mammalian models for the targeted delivery of growth factors, such as BDNF and NT-3 [150,151]. The comparative ease of genetic manipulation in zebrafish, combined with these advanced genetic tools in mammalian models, offers a comprehensive platform for cross-species studies. This synergistic approach might be pivotal for translating the promising findings from zebrafish into potential therapies for mammalian SCIs.

A main advantage of using a transgenic zebrafish model for SCI research is the considerably shorter generation time and higher fecundity compared to mammals [152]. This allows for a more rapid generation of transgenic lines and, consequently, faster data collection. Additionally, zebrafish embryos are transparent and develop externally [153], facilitating in vivo imaging studies to track neural regeneration and other cellular events in real time [154]. Another significant benefit is the reduced ethical considerations and costs associated with zebrafish research compared to a mammalian model. The use of zebrafish allows for a more straightforward ethical approval process and is often less resource-intensive, making it more accessible for many research groups. Furthermore, zebrafish have a remarkable ability to regenerate their spinal cord tissue, which is not commonly found in mammalian model [155]. This provides a unique opportunity to study successful neural regeneration processes that can then be translated into the mammalian model for therapeutic approaches. However, it is essential to note that each model system has its own set of advantages and limitations. While zebrafish offers high-throughput screening and regenerative capabilities, the mammalian model provides a closer physiological relevance to human conditions [156]. Therefore, the use of both zebrafish and mammalian models can offer complementary insights into the complex processes of SCIs and neural regeneration.

In conclusion, the zebrafish model system offers a substantial potential to further our understanding of neural regeneration following SCIs. We also proposed a methodology for conducting research on nervous system regeneration using the zebrafish as an animal model (Figure 2). By exploring the complex cellular and molecular interactions within zebrafish, researchers can elucidate the vital mechanisms, signaling pathways, proteins, and cell types involved in neural regeneration. This knowledge can lead to the development of innovative therapeutic strategies to improve neural regeneration and functional recovery in humans and other mammals affected by SCIs.

## 4. Future Directions: Zebrafish Neuron Regeneration Research Informs Mammalian SCI Therapy Development

### 4.1. Insights into and Therapeutic Potential of Zebrafish Models in SCI Research

The zebrafish model offers a unique opportunity to study neural regeneration after an SCI, providing insights into the cellular and molecular mechanisms, as well as potential therapeutic targets [157]. Advancing our understanding of neural regeneration in SCIs requires overcoming several key challenges, including the identification of precise cell types involved, addressing the abnormal microenvironment at the injury site, tackling the inhibitory immune response, managing glial scar formation, and addressing large gaps at the injury site due to severe SCIs. However, even with axonal regeneration, functional recovery remains limited due to the challenges in integrating regenerative axons into existing neural circuits and reestablishing functional synaptic connections (Figure 1). Zebrafish studies have already yielded valuable information on potential candidate proteins involved in neuroregeneration following SCIs, such as CTGFa [132], Cav1 [51], ANP32a [133], matrix metalloproteinase-9 [158], and sonic hedgehog [159,160]. Furthermore, in the current zebrafish study, differentially expressed genes were sorted and analyzed in-depth using bioinformatics methods, such as the Notch signaling pathway [161,162], Wnt signaling pathway [163], and Hippo-Yap/Taz signaling pathway [164]. These results suggest that, after an SCI in adult zebrafish, our understanding of the molecular mechanisms underlying axon regeneration can be facilitated, and these candidate genes and pathways can serve as therapeutic targets for the treatment of CNS injuries. 

Additionally, the zebrafish model is allowed for the manipulation of known or unknown genes and proteins, which could be applied to mammalian SCI models. For instance, BDNF is a protein involved in the survival and growth of neurons, as well as synaptic plasticity [165,166]. It has been shown to play a role in axonal regeneration and functional recovery in zebrafish following SCIs [78,167]. The overexpression of BDNF in zebrafish has been found to improve axonal regeneration and locomotor recovery [168,169], suggesting that promoting BDNF expression or signaling could be a potential therapeutic target for mammalian SCI treatment. Similarity, fibroblast growth factors (FGFs) are a family of proteins involved in various cellular processes, including cell growth, differentiation, and tissue repair [170]. In zebrafish, FGF signaling was implicated in the regenerative response following SCIs. Enhancing FGF signaling has been shown to promote axonal regeneration, reduce glial scarring, and improve locomotor recovery in zebrafish models of SCIs [171,172]. This finding suggests that targeting FGF signaling might also be beneficial for mammalian SCI treatment. While the regenerative capacities of zebrafish and mammals differ, understanding the molecular mechanisms that promote axonal regeneration and functional recovery in zebrafish can provide valuable insights into developing novel therapeutic strategies for mammalian SCI treatment.

The availability of various transgenic lines in zebrafish helps to unravel the complex cell types involved in neural regeneration during SCIs. However, the identification of cell populations involved in nerve repair is challenging due to their heterogeneity and the limitations of single marker proteins or specific gene transgenic strains. Furthermore, it is crucial to address the abnormal microenvironment and inhibitory immune response that hinder nerve regeneration and cell survival at the injury site. As we continue to study the zebrafish model and explore its high regenerative potential, we can expect to gain more insights into the molecular mechanisms, signaling pathways, proteins, and cell types engaged in neural regeneration. This knowledge will help develop novel therapeutic strategies and promote neural regeneration and repair in mammals suffering from SCIs. The future of SCI research using zebrafish models will rely on intensive study, addressing the current challenges, and translating the findings.

### 4.2. Using Zebrafish to Identify Therapeutic Targets in SCI Research

The zebrafish has emerged as a powerful model organism in biomedical research, particularly in the field of neural regeneration and SCIs. Its unique biological characteristics and genetic tractability have positioned the zebrafish as an invaluable resource for exploring complex biological processes, including those underlying SCIs. The insights gained from zebrafish studies have not only enriched our understanding of neural regeneration mechanisms, but they have also paved the way for innovative therapeutic strategies. The zebrafish model offers several unique advantages that make it an excellent tool for identifying novel therapeutic targets.

#### 4.2.1. Regenerative Capacity

Zebrafish has a remarkable ability to regenerate their nervous system, including the spinal cord. Studying the inherent mechanisms of regeneration in zebrafish can lead to the identification of critical genes, proteins, and signaling pathways that can be targeted in therapeutic interventions for mammals [50,173]. This aspect has been extensively studied, revealing insights into the genetic and cellular factors that enable regeneration.

#### 4.2.2. Genetic Manipulability

Zebrafish is amenable to genetic techniques, such as CRISPR/Cas9. For instance, Keatinge et al. (2021) used synthetic CRISPR guide RNAs to target macrophage-related genes, identifying key regulators, such as tgfb1a, which affected spinal cord regeneration [174]. This highlighted the utility of zebrafish for understanding complex biological processes and identifying therapeutic targets.

In summary, the zebrafish model provides a multifaceted platform for investigating neural regeneration. Its regenerative capacity, genetic manipulability, transparency, suitability for high-throughput screening, and conserved molecular pathways collectively enabled the identification of novel targets for therapeutic intervention in SCIs [175]. We believe that continued research using the zebrafish model will undoubtedly contribute to innovative treatments for SCIs in the future.

### 4.3. Challenges and Limitations of the Zebrafish Model in SCI Research

The zebrafish model has emerged as an essential tool in SCI research, providing key insights into neural regeneration, therapeutic interventions, and the underlying molecular mechanisms. However, alongside its numerous advantages, certain challenges and limitations must be acknowledged. We highlighted three key areas that illustrate both the limitations and constraints of utilizing a zebrafish model in SCI research.

#### 4.3.1. Anatomical and Physiological Differences

The zebrafish shares many genetic similarities with humans; however, differences in anatomy and physiology may limit the direct translatability of the findings. For example, the ability of zebrafish to regenerate whole organs, such as the heart [176], is a feature not present in mammals, and this significant difference in the regenerative capacity may pose challenges in translating the discoveries from zebrafish to human therapies. Complementary studies on mammalian model may be needed to validate the results derived from zebrafish.

#### 4.3.2. Genetic Complexity

Although genetic manipulation is feasible in zebrafish, it can be challenging and time-consuming, particularly for intricate genetic studies. For example, while targeted gene knockouts can be achieved using CRISPR/Cas9 technology, generating multiple knockouts or complex genetic constructs may require an extensive optimization that can be labor-intensive. Understanding the interactions between multiple genes or pathways in zebrafish may thus require significant investments in time and resources. The continued advancement of genetic methodologies will enhance this model’s utility.

#### 4.3.3. Limited Adult Brain Models

Much zebrafish research focuses on embryos or larvae, which may not fully represent adult human brain and spinal cord complexities. For example, while embryonic zebrafish model provides valuable insights into neural development and regeneration, they might not capture the age-related changes in neural plasticity or the response to injury that occurs in the adult human CNS. The development of adult zebrafish SCI models that better mimic the complexity of the adult human nervous system can bridge this gap and allow for more accurate extrapolations to human conditions.

In summary, while the zebrafish model offers a versatile and valuable platform for SCI research, recognizing and strategically addressing these limitations is the key to its continued success. Collaborations across disciplines, the refinement of experimental techniques, and a focus on the translational research are essential to leverage the full potential of the zebrafish model in advancing SCI therapeutics.

## 5. Conclusions

In conclusion, this review highlighted the significance of a multi-faceted approach to understanding and addressing SCI and its treatment. By examining the research and advancements made concerning different species, including rodents, non-human primates, and zebrafish, we gained a broader perspective on the underlying cellular and molecular mechanisms of SCIs and the potential therapeutic interventions. To develop effective therapies, it is vital to have a comprehensive understanding of the complex interplay between various cell types, molecular pathways, and signaling mechanisms that play major roles in neural repair and regeneration. Cell-based therapies have shown their potential in the preclinical studies; however, their translation to clinical settings is hindered by our limited knowledge of the specific cell populations involved in neural regeneration and the challenges posed by the SCI microenvironment. These challenges include inflammation, glial scarring, hypoxia, and the vulnerability of certain cell types to stress.

Emerging strategies, such as tissue engineering and gene therapy, show promise in overcoming these obstacles and promoting axonal regeneration and functional recovery after an SCI. By combining novel approaches with advanced biomaterials and targeted gene delivery systems, researchers are working towards developing innovative solutions to enhance neural repair and regeneration. Furthermore, the future of SCI research relies on interdisciplinary collaborations, integrating expertise from various fields, such as neurologic medicine, immunology, molecular and cellular biology, genetics, structural biology, material science, and computer science and engineering. By fostering these collaborations, researchers can overcome the complex challenges associated with SCIs and work towards the development of more effective therapies. As we continue to refine our understanding of the intricate processes involved in SCIs and neural regeneration, we are better equipped to develop therapeutic interventions that can significantly improve the quality of life and functional outcomes for patients suffering from SCIs.

## Figures and Tables

**Figure 1 ijms-24-13938-f001:**
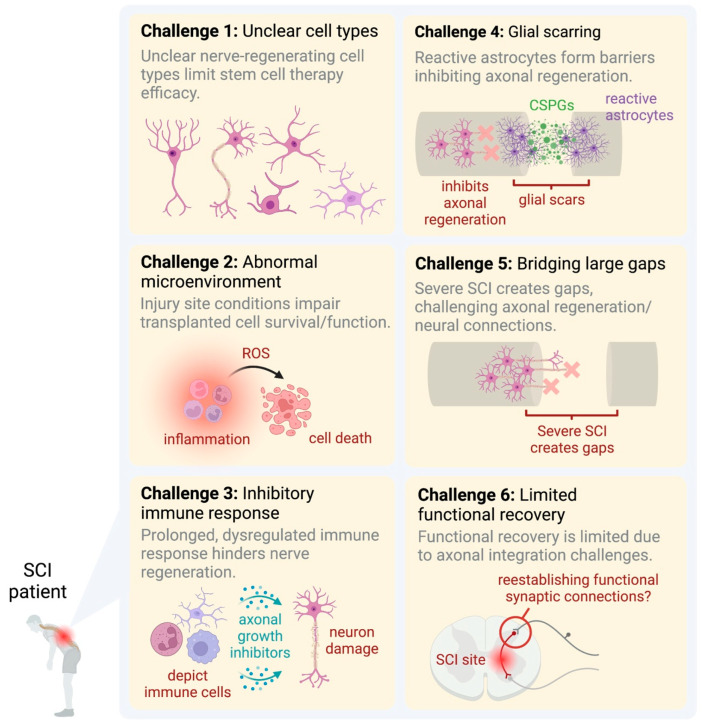
Comprehensive illustration of the six major challenges in SCI treatment. Challenge 1: unknown cell types involved in nerve regeneration make it difficult to optimize stem cell transplantation therapies. Challenge 2: the harsh microenvironment at the injury site, characterized by inflammation, cell death, and oxidative stress, negatively impacts the survival and function of transplanted cells. Challenge 3: the inhibitory immune response, initially beneficial, often becomes prolonged and dysregulated, leading to further tissue damage and hindering nerve regeneration. Challenge 4: glial scarring, caused by reactive astrocytes and the accumulation of CSPGs, establishes a physical and chemical barrier that inhibits axonal regeneration. Challenge 5: in cases of a severe SCI, large gaps or cavities form at the injury site, posing a significant challenge for axonal regeneration and the re-establishment of neural connections. Challenge 6: despite achieving axonal regeneration, limited functional recovery remains a major issue due to the difficulty in integrating regenerated axons into existing neural circuits and reestablishing functional synaptic connections.

**Figure 2 ijms-24-13938-f002:**
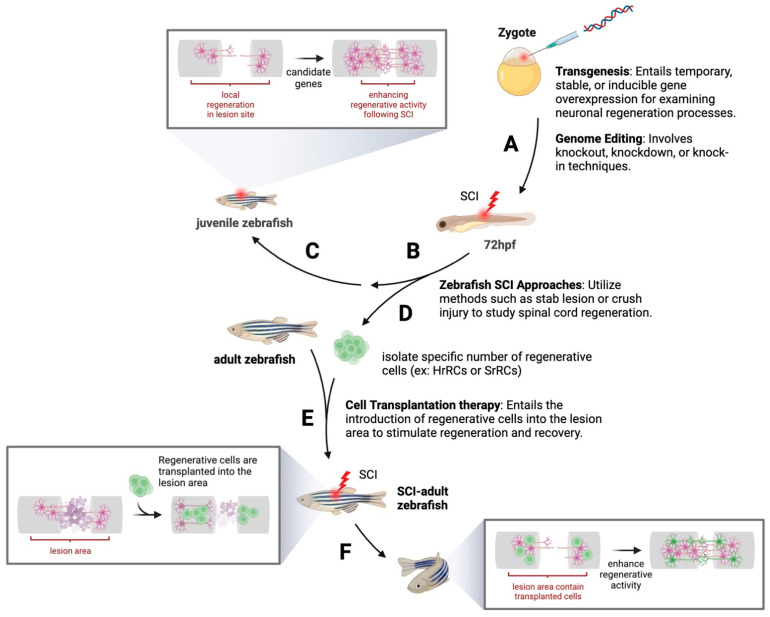
This figure presents a comprehensive schematic of a zebrafish experimental model utilized in nervous system regeneration research. It illustrates a multi-step process that can be broken down into the following stages. (**A**) It begins with the identification and injection of a regenerative candidate gene into a zebrafish zygote. This intricate process leverages transgenesis and genome editing techniques to manipulate gene expression, thereby influencing subsequent developmental and cellular processes. (**B**) Subsequent to the genetic manipulation, spinal cord injury (SCI) is performed in the zebrafish embryos using well-established methods, such as the stab lesion or crush injury described in this text. This approach serves as an excellent model for studying the biological and molecular mechanisms of spinal cord regeneration. (**C**) The next phase involves a rigorous investigation to determine whether the input candidate gene promotes the regenerative activity following an SCI. This is accomplished through a series of in vivo experiments, which provide insights into the gene’s potential therapeutic implications. (**D**) As part of the study, specific numbers of regenerative cells, such as hypoxia-responsive recovering cells (HrRCs) or SCI stress-responsive regenerating cells (SrRCs) described in this text, can be particularly isolated. This step allows for a closer examination of the cells’ morphological characteristics, behavior, migration, and regenerative capacity. (**E**) These isolated regenerative cells are then transplanted into another adult SCI-treated zebrafish. This step allows for the investigation of the potential therapeutic benefits of cell transplantation in promoting recovery after an SCI. (**F**) Finally, the model allows for the observation and assessment of whether the transplanted cells, which harbor the examined candidate gene initially injected into the zygote, can improve neuronal regeneration in the SCI-affected zebrafish. This comprehensive model provides a robust and versatile platform for exploring the complex mechanisms of neuronal regeneration, with the ultimate goal of identifying potentially effective therapeutic strategies for SCI treatment.

**Table 1 ijms-24-13938-t001:** Examples of transgenic zebrafish lines and key findings for studying neural regeneration after an SCI and their implications for mammals.

Transgenic Line/Technique	Purpose and Function	Key Finding	Implications for Mammalian Neural Regeneration	References
*Tg(her4.1:mCherryT2ACreERT2)*	label *her4.1*-positive ventricular radial glial progenitor cells	identified radial glial progenitor cells in regeneration	understanding the role of progenitor cells	[52]
*Tg(−3.5dbx1a:egfp)*	label radial glial progenitor cells differentiating into neurons	identified cells differentiating during embryogenesis	insights into cell differentiation in mammals	[53]
*Tg(−8.4ngn1:egfp)*	label young migrating neural progenitor cells (NPCs)	tracked NPC migration in regeneration	study migration patterns of progenitor cells	[104]
*huORFZ* ^a^	ER stress-responsive subtypes displaying GFP expression	identified stress-responsive recovering cells (ex: HrRCs ^b^, SrRCs ^c^)	understanding stress-induced regeneration (ex: hypoxia, and mechanical injury)	[28,43,108]
Candidate gene/protein study	identify proteins involved in neuroregeneration following SCI	connective tissue growth factor a, Caveolin 1, ANP32a	identify potential targets for therapy	[33,43,105]
Neuropeptide Y	manipulate gene/protein expression in zebrafish and mammalian models	neuropeptide Y promotes axonal regeneration	developing novel therapeutic approaches	[109,110]

Note: ^a^ This zebrafish transgenic line contains a unique motif called human *uORF^chop^* (*huORF^chop^*) to regulate the translation of a downstream coding sequence within mRNA; ^b^ hypoxia-responsive recovering cells; ^c^ SCI stress-responsive regenerating cells.

## Data Availability

Not applicable.

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
