# Peer review of "The Promising Role of a Zebrafish Model Employed in Neural Regeneration Following a Spinal Cord Injury"

_ijms, 2023, doi:10.3390/ijms241813938_

Round 1

Reviewer 1 Report

needs a through revision for better readability 

Reviewer 2 Report

The review presented by the authors is very interesting, however, in order to improve the manuscript, several changes must be introduced in the final version:

INTRODUCTION

In recent decades, many articles have been published in which stem cells have been used as a method of axonal repair and regeneration after spinal cord injury in animal models. In addition to the articles cited in the manuscript, the authors should include information on stem cell transplants in human subjects with spinal cord injury, which allows corroborating that this therapy has clinical promise. Please include these stem cell transplant clinical studies in the final version of the manuscript.

In mammals, the regeneration capacity of injured axons after a spinal cord injury depends mainly on the environment that is generated around injury and adjacent regions. Classical studies by Professor Aguayo (1980; show that central neurons have the capacity to regenerate when this environment is favorable (Nature. 1980 Mar 20;284(5753):264-5). Building on these pioneering studies, many investigators have used various strategies to promote the growth of injured axons through the area of injury. Authors should include information on these strategies used to promote central regeneration in mammalian experimental models, and not just focus on studies of neuro-spheres. Please include this relevant information in the final version of the manuscript.

Spinal cord injury results in glial scarring that limits axonal growth. As the authors indicate, microglia and astrocytes are the main cellular elements of this glial scar, which secrete proteoglycans and other molecules that inhibit axonal growth. In the scientific literature, at least two strategies have been described to promote axonal growth by glial scarring: one is the use of chondroitinase ABC, and the second is the use of ensheathing glial cells of the olfactory bulb or olfactory mucosa, that generate "channels" through which regenerative axons can pass. The authors should include in the final version of the manuscript the strategies that have been used to promote axonal growth around injury where the glial scar is generated. Please include this relevant information in the final version of the manuscript.

CURRENT CHALLENGES IN THE TREATMENT OF SPINAL CORD INJURY

Currently, the only treatment that has worked in a patient with incomplete spinal cord injury has been transplantation of olfactory ensheathing glial cells together with peripheral nerve grafts (Cell Transplant. 2014;23(12):1631-55). If the authors know of more clinical studies where any of the strategies indicated in the introduction section have worked, they should also include them in the manuscript. This information is relevant and reinforces the authors' message about the scarcity of functional reparative therapies in humans.

In point 2.1. the authors focus only on stem cell transplantation therapy. Prior to stem cell studies, many other researchers have used other cells to promote regeneration in the injured spinal cord. Why focus the review only on stem cells? Most studies show that 90% of stem cells differentiate into glial cells such as astrocytes, and only 10% of transplanted stem cells differentiate into neurons. Very few studies show that stem cells differentiate into neurons and that these new neurons generate functional neural circuits. It is important that the authors clarify why they are only reviewing stem cell transplantation. On the other hand, there are more than 25 previous articles that have also reviewed the role of stem cell transplants after spinal cord injury. What new and relevant information does this manuscript provide compared to previous reviews? 

In point 2.2. the authors provide evidence that there is a microenvironment in the lesion area that does not favor the survival of the transplanted cells. However, many authors who use the cell transplant paradigm to promote regeneration in the spinal cord show that the transplanted cells do survive even in this adverse environment. Please include these articles in which cell transplant survival after spinal cord injury is demonstrated and discuss these results with the evidence indicated by the authors in the manuscript. Include all this information in the final version of the manuscript.

In point 2.3. and point 2.4, the authors should include other cell types that have also been used in animal models of spinal cord injury, and that also reduce inflammation. Not only stem cells have this role in controlling gliosis and post-injury inflammation of the spinal cord. The authors should include more previous studies from the scientific bibliography in the final version of the manuscript.

In point 2.5., there are more and more studies that analyze the biocompatibility of the materials used, as well as their physical properties. Authors should include information from these previous studies in the final version of the manuscript.

In point 2.6., the authors should include information about the article where it is shown that the neurons generated from stem cell transplantation do integrate into spinal cord circuits, giving functional synapses. Likewise, at this point the authors should explain in more detail the strategies they indicate, such as electrical stimulation, optogenetic manipulation and administration of neuro-modulatory agents. What relevant results have been obtained from these strategies? Please include all this relevant information in the final version of the manuscript.

ZEBRAFISH PRESENT AN OPPORTUNITY TO STUDY NEURAL REGENERATION IN SCI

In this section on zebrafish, it is very important to begin with a methodological section on spinal cord injury models in zebrafish. What is the method used to perform compression and contusion of the zebrafish spinal cord? Are these lesion types in zebrafish comparable to the same lesion types in mammals? Are the pathophysiological changes observed in these types of lesions in zebrafish like those observed in mammals? All this information is very relevant, and the authors should include it as a first point in this section.

The information included in Table 1 is very interesting. Genetic tools only have been used in the zebrafish model to study axonal regeneration after SCI? these tools have been used in mammals with spinal cord injury? Please clarify these points and include this information in the final version of the manuscript.

Transgenic lines are very interesting for studying pathophysiological processes, including those that occur after spinal cord injury. The authors give evidence of different transgenic lines in the zebrafish model. However, transgenic animals have also been used in mammals to study what happens after spinal cord injury. Why is it better to use transgenic zebrafish than transgenic mammals such as mice? Please clarify this point and include this information in the final version of the manuscript.

There are previous reviews on the zebrafish model and spinal cord injury (Mol Biol Rep. 2021 Feb;48(2):1787-1800; Front Mol Neurosci. 2022 Sep 7;15:983336; Curr Opin Genet Dev. 2020 Oct;64:44-51; Cells. 2021 Jun 6;10(6):1404; Dis Model Mech. 2020 May 27;13(5):dmm044131; IBRO Neurosci Rep. 2023 Apr 3;14:441-446; Biomedicines. 2021 Jan 12;9(1):65; Heliyon. 2020 Feb 29;6(2):e03507; Rev Neurosci. 2018 Dec 19;30(1):45-66; Neural Plast. 2016;2016:5815439; Int J Mol Sci. 2023 Mar 30;24(7):6483; Brain Res Rev. 2008 Jan;57(1):86-93; Restor Neurol Neurosci. 2008;26(2-3):71-80). What new and relevant information does this manuscript provide regarding all these previous reviews? Please, the authors should clarify this point and include this information in the final version of the manuscript.

The title of the manuscript is "The Promising Roles of Zebrafish Model Employed in Neural Regeneration after Spinal Cord Injury", and the authors provide very little relevant information on the promising roles of the zebrafish model in the context of spinal cord injury and axonal regeneration. The authors do not explain the pathophysiology of spinal cord injury in zebrafish, nor the therapies that have been used to promote axonal regeneration in zebrafish after spinal cord injury. The first two sections are information obtained in a mammalian model, but not from zebrafish. The authors also do not make a comparison between the zebrafish model and the mammalian model after spinal cord injury, explaining the pros and cons of both models, and highlighting the advantages of the zebrafish model over the mammalian model. The authors do not explain which genes are relevant for axonal regeneration after spinal cord injury in the zebrafish model, which makes these animals have greater regenerative success. They do not explain whether these genes and the proteins derived from these pro-regenerative genes have been conserved in evolution, and which ones have been lost causing the regenerative capacity of the central axons to be limited in mammals, but not in fish such as the zebra fish. These are all points that the authors could have addressed in this review. What is the originality of this manuscript with respect to previous articles on a similar theme?. 

Round 2

Reviewer 1 Report

Can be accepted 

Minor changes required 

Reviewer 2 Report

The authors have provided very satisfactory answers to the reviewer's questions and comments, and have included new information in the final version of the manuscript, which has significantly improved this latest version.